# Outcomes of Temporary Hemiepiphyseal Stapling for Correcting Genu Valgum in Children with Multiple Osteochondromas: A Single Institution Study

**DOI:** 10.3390/children8040287

**Published:** 2021-04-08

**Authors:** Giovanni Trisolino, Manila Boarini, Marina Mordenti, Andrea Evangelista, Giovanni Gallone, Stefano Stallone, Paola Zarantonello, Diego Antonioli, Giovanni Luigi Di Gennaro, Stefano Stilli, Luca Sangiorgi

**Affiliations:** 1Unit of Pediatric Orthopedics and Traumatology, IRCCS Istituto Ortopedico Rizzoli, 40136 Bologna, Italy; giovanni.trisolino@ior.it (G.T.); giovanni.gallone01@gmail.com (G.G.); stallone.stefano@gmail.com (S.S.); p.zarantonello1@gmail.com (P.Z.); diego.antonioli@ior.it (D.A.); giovanniluigi.digennaro@ior.it (G.L.D.G.); stefano.stilli@ior.it (S.S.); 2Department of Rare Skeletal Disorders, IRCCS Istituto Ortopedico Rizzoli, 40136 Bologna, Italy; manila.boarini@ior.it (M.B.); andrea.evangelista@ior.it (A.E.); luca.sangiorgi@ior.it (L.S.)

**Keywords:** multiple osteochondromas, genu valgum, hemiepiphyseal stapling, growth modulation, children, knee

## Abstract

Background: Multiple osteochondromas is a rare skeletal disorder characterized by the presence of osteocartilaginous protrusions causing bony deformities, especially around the knee. Guided growth by temporary hemiepiphyseal stapling is the treatment of choice to correct the deformity by modulating the residual physeal growth of the lower limbs. Although this procedure is increasingly practiced, inconclusive evidence exists regarding its effectiveness in children with multiple osteochondromas. The study aims to compare the outcomes of temporary hemiepiphyseal stapling for correcting genu valgum in children with multiple osteochondromas vs. idiopathic cases. Methods: In this retrospective cohort study, we included patients admitted at a single institution from 2008 to 2018. A total of 97 children (77 idiopathic, 20 multiple osteochondromas) were enclosed, accounting for 184 limbs treated by temporary hemiepiphyseal stapling. We investigated if children with multiple osteochondromas had a similar successful rate of correction, rate of complications, and correction velocity compared to children with idiopathic genu valgum. Results: Overall, 151 limbs (82%) achieved complete correction or overcorrection, with idiopathic cases having a significantly higher rate of success compared to pathologic cases (88% vs. 55%; *p* < 0.001). In addition, multiple osteochondromas children sustained a higher rate of major complications (*p* = 0.021) and showed significantly lower correction velocity (*p* = 0.029). Conclusion: Temporary hemiepiphyseal stapling is effective in both idiopathic and multiple osteochondromas children, although the latter often achieved incomplete correction, had a higher risk of complications, and required a longer time of stapling. We suggest to anticipate the timing of intervention; otherwise, children with multiple osteochondromas and severe valgus deformity, approaching skeletal maturity, could undergo combined femoral and tibial stapling.

## 1. Introduction

Multiple osteochondromas (MO; OMIM #133700, #133701), is a rare condition caused by mutations on the exostosin-1 (EXT1) or exostosin-2 (EXT2) gene in 90% of patients [1]. The MO signature feature is the presence of osteocartilaginous protrusions, named osteochondromas (OCs), that mainly arise from physes of long bones and grow until skeletal maturity [2,3]. MO frequently leads to pain, joint limitations, and bony deformities [4,5]. MO patients are known to have a high rate of knee deformities due to the disarrayed growth of the physes [6,7]. Genu valgum is the most common knee deformity encountered in MO, present in almost 38% of cases [8] and is also frequently encountered in the general pediatric population [9,10]. The physiological deformities are treated conservatively, but severe genu valgum, causing pain, instability, functional limitations, and cosmetic concerns, requires correction [11,12]. Recently, guided growth by temporary hemiepiphyseal stapling (HeS) has become the treatment of choice to correct the knee deformities by modulating the residual growth of the physes [12,13,14,15,16,17]. To date, there is incomplete evidence concerning the effectiveness of HeS in pathologic genu valgum, especially in children with genetic skeletal disorders, such as MO, and pathologic physes [15,18,19,20]. Therefore, we aimed to investigate the outcomes of HeS in MO patients compared with children with idiopathic genu valgum. We asked if MO children have a similar rate of correction, rate of complications, and correction velocity compared with children with idiopathic genu valgum and analyzed which factors could predict the rate and velocity of correction.

## 2. Materials and Methods

### 2.1. Study Design and Participants

We conducted a retrospective cohort study in children with idiopathic or pathologic genu valgum due to MO, admitted at a single institution from 2008 to 2018. Our institution was a referral center that is highly specialized in rare skeletal disorders.

The hospital database was queried for ICD-9 codes (736.41, 755.64, 755.69, 736.89, 755.63); in addition, the Registry of Multiple Osteochondromas (REM-NCT04133285), a standardized tool capturing MO patient longitudinal data from both our institution and external healthcare providers (HCP), was queried for Orphanet code (ORPHA:321). Overall, 4013 patients were screened for eligibility (Figure 1).

The inclusion criteria were as follows: male and female <18 years; valgus angular deformity of the knee on the coronal plane, as identified clinically by an inter-malleolar distance (IMD) higher than 8 cm and confirmed radiographically by the hip–knee–ankle angle (HKA) < −2° [21,22]; presence of open physes at stapling; hemiepiphyseal stapling at distal femur and/or proximal tibia using eight-plate treatment; availability of high-quality full weight-bearing long-standing radiographs taken preoperatively and at the time of staple removal. All inclusion criteria had to be satisfied to be enrolled in the present study.

The exclusion criteria were as follows: children sustaining surgical treatments other than hemiepiphyseal stapling (e.g., osteotomies, circular external fixation, etc.); patients with genetic skeletal disorders other than MO (e.g., Ollier disease, multiple epiphyseal dysplasia, etc.) [23]; children with acquired deformities, such as post-traumatic, metabolic, neuromuscular, or sequelae of surgical interventions; furthermore, we excluded 10 children undergoing HeS by means of Blount staples, since they were progressively abandoned in our practice and could have different correction velocity, compared to the eight-plate treatment.

The study protocol was approved by the institutional review board (ID: 735/2020/Oss/IOR—n. 10820/2020) according to Italian regulations (art. 110bis of D.lgs. n.196/2003). Explicit consent was not required for retrospective chart reviews and publication of the aggregated data.

The study was conducted following the most updated version of Declaration of Helsinki (Fortaleza, October 2013), as well as all the national and international law for clinical research. The protocol was written according to the International Conference on Harmonization guidelines for Good Clinical Practice (ICH-GCP) provided by the European Medicines Agency.

### 2.2. Surgical Procedure

Surgical procedure was planned and performed by 11 surgeons belonging to the Unit of Pediatric Orthopedics and Traumatology, following a standardized surgical technique.

Under general anesthesia, the patient was placed in supine position on a radiolucent operating table with a thigh tourniquet. Femoral and/or tibial growth plates were identified under fluoroscopy using a needle as a landmark. 

A 2–3 cm long incision was made, centered on the physes in each plane, over the medial distal femur or the medial proximal tibia. For the femur, the dissection was carried out through the fascia of the vastus medialis. In both approaches, the leash of epiphyseal vessels was visualized, leaving the periosteum undisturbed, to prevent physeal bar formation and physeal arrest. 

Eight plates were placed centered over the physes under fluoroscopic guidance. The implant size varied between 24 and 32 mm for the screws and 12 and 16 mm for the plate (Eight-Plate Guided Growth System™, Orthofix Medical Inc. Lewisville, TX, USA). Layered closures were then performed.

Postoperatively, children could completely mobilize the knee and walk with full weight-bearing and they were usually discharged by the second or third day after surgery. Regular clinical assessment occurred postoperatively at 1, 3, and 6 months, checking the correction of the deformity. When the corrected alignment was reached or a slight overcorrection was achieved, the patient was evaluated with long-standing radiographs and hardware removal was performed.

### 2.3. Assessment of Baseline Variables and Outcomes

Clinical variables and radiographic parameters were collected preoperatively and at staple removal. Demographic variables included gender and age at stapling. Anthropometric parameters included body weight, height, body mass index (BMI), and related z-scores, based on the Italian reference charts [24]; in particular, a BMI z-score ranging between 1 and 2 was associated with being overweight and values >2 were associated with obesity. The IMD was measured with the medial side of the knees in light contact and the patella pointing forwards; clinically, the axis of the knee was considered neutral with an IMD ranging from 0 to 2 cm [10]. Surgical variables included laterality, the site and side of stapling, type of hardware, duration, and correction velocity. Radiographic parameters were calculated on long-standing anteroposterior radiographs of the lower extremities, digitalized on pictures archiving and communication system (PACS). Radiographic parameters included the following: mechanical hip–knee–ankle angle (HKA) and mechanical axis deviation (MAD), both calculated as the absolute value and as a percentage to one-half of the width of the tibial plateau [18]; medial proximal tibial angle (MPTA); anatomical and mechanical lateral distal femoral angle (aLDFA and mLDFA, respectively); the length of the femur, tibia, and fibula, as well as the tibia-fibula ratio (tibial length/fibular length) and femur-tibia ratio (femur length/tibial length) [25,26] (Appendix A and Figure 2A,B).

The reliability of radiographic measurements was tested by five independent blinded observers and was found to be good to excellent (Appendix A). 

According to a method proposed by several authors [18,19], we identified four zones on both the medial and lateral side of the knee, on the basis of the MAD location at staple removal (−2 or less = uncorrected; −1 = partially corrected; +1 = corrected; +2 or more = overcorrected). Patients achieving correction or overcorrection were considered as a successful outcome. The correction velocity was also calculated as the variation of MAD from baseline to staple removal (mm/months). Complications were rated according to the modified Clavien-Dindo-Sink complication classification system [27,28] (Appendix A). Adverse effects graded >2 were considered as major complications. 

### 2.4. Statistical Analyses

Patients and limbs characteristics were compared between MO and idiopathic groups using the Mann–Whitney test and χ^2^ or exact Fisher test, if appropriate, for continuous and categorical variables, respectively. Factors affecting mean differences (MD) on postoperative MAD and correction velocity were evaluated using linear regression models. Because of the reduced number of events, the proportion of adverse effects graded >2 was compared between the MO and idiopathic groups using a logistic regression model and adjusting with a propensity score (PS) that was estimated on the basis of the patients’ and limbs’ characteristics. In all models, the effects were adjusted for preoperative variables. To account for clustering of limbs within patients, standard errors were estimated using the clustered sandwich estimator. Main analyses were performed by applying regression models on observations with complete data (complete-case analysis approach). In addition, the results’ consistency was evaluated by adjusting the comparison between the two populations by applying different propensity score approaches [29]. A *p*-value < 0.05 was considered statistically significant. All analyses were performed with Stata 11.2 (StataCorp^®^, College Station, TX, USA).

## 3. Results

### 3.1. Patients Characteristics at Baseline

Overall, 97 children (77 idiopathic, 20 MO) were included (Table 1). 

As expected, the two groups differed for age at surgery and for anthropometric measurements [15,19]; in particular, the MO cohort received surgery at an early age (about one year before the idiopathic cases) and showed shorter stature and lower BMI. Children with idiopathic genu valgum underwent bilateral stapling in 96% (74/77) of cases compared to MO patients, who had bilateral stapling in 65% (13/20) of cases (*p* < 0.001). Overall, we evaluated 184 limbs (151 idiopathic, 33 MO; Table 2 and Table 3). 

We observed that MO children had a higher radiographic angular deformity at baseline, despite that the malalignment appeared similar from a clinical point of view (IMD was comparable between groups; Table 1). On radiographs, the MPTA differed significantly between the two groups, as well as the aLDFA. Furthermore, we observed significant difference in the tibia-fibula ratio and femur-tibia ratio (*p* < 0.001), meaning that children with MO had a slight hypoplasia of the tibia and fibula.

The site of stapling involved mainly the distal femur in idiopathic cases (120/151, 79.2%), while children with MO were mostly stapled at proximal tibia (21/33, 63.6%). Combined femoral and tibial stapling was performed in 5.4% (10/184) of limbs. 

### 3.2. Clinical and Radiographical Outcomes

Pre- and postoperative IMD was available in 86 children. In total, 53 idiopathic patients out of 70 (76%) and 10 MO patients out of 16 (63%) achieved complete clinical correction or slight overcorrection at staple removal.

Radiographically, 151 limbs out of 184 (82%) achieved complete correction or overcorrection, which got gradually corrected over the follow-up period, with idiopathic cases having a significant higher rate of success (88% vs. 55%; *p* < 0.001) and faster correction (1.66 mm/month vs. 0.93 mm/month, *p* < 0.001). In total, 6 limbs out of 184 (3.3%) failed to achieve correction; furthermore, 27 limbs (14.7%) achieved only partial correction. Detailed results are reported in Table 3. 

Multivariable linear regression analysis confirmed that MO (β-coefficient = −0.73; *p* = 0.03), male gender (β-coefficient = 0.40; *p* = 0.05) and preoperative MAD (β-coefficient = 0.02; *p* < 0.0001) were independent predictors of correction velocity, while height, adjusted for age and gender (β-coefficient = 4.11; *p* = 0.002), and the preoperative MAD (β-coefficient = 0.41; *p*-value < 0.0001) influenced the rate of correction (Appendix A and Figure 3A,B). Major complications in MO children included supplementary stapling for valgus rebound (4/184) and surgery for recurrent OCs around the knee (6/184); on the other hand, major complication in idiopathic patients requiring further surgical interventions included the following: one deep infection following staple removal that was treated by surgical debridement and 2-week antibiotic therapy, one plate malpositioning that was treated by plate repositioning 2 months after the first operation, and one tibial overcorrection requiring valgus osteotomy at the end of growth. MO children sustained a higher rate of major complications and further surgical procedures (30% vs. 2%, *p* < 0.001); this difference remained statistically significant even after excluding subsequent interventions for OCs’ excision at the same site of stapling (12% vs. 2%, *p* < 0.001). Detailed results are reported in Table 3.

The results were consistent also applying different propensity score approaches (See Appendix A).

Two illustrative cases are showed in Figure 4A–D.

## 4. Discussion

The study investigated one of the largest case series of HeS for correcting genu valgum in MO children in comparison with idiopathic cases [12,15,16,17,18,19]. We confirmed that temporary HeS is safe and effective in correcting genu valgum during growth, achieving complete clinical and radiographic correction in more than 80% of knees. Nonetheless, we demonstrated that MO is an independent predictor of reduced correction velocity and a higher rate of complications. Moreover, MO seems to be a predictor of incomplete correction, although, with the numbers available, it was borderline not statistically significant. This is in line with previous studies that reported limited effectiveness of guided growth in case of pathologic physes [15,20]. Boero et al. reported a 79% rate of satisfactory correction in a cohort of 28 patients with pathologic physes and mixed deformities, compared to a 100% rate of complete correction in a cohort of 30 children with idiopathic deformities. Wiemann et al. reported a significantly higher complication rate in children with abnormal physes (28% vs. 7% of complications in idiopathic cases) in their cohort of 38 children [20]. In contrast, Kang et al. investigated a cohort of 11 MO children, compared to 33 idiopathic cases, and confirmed that correction velocity was slower in MO children, although they observed a comparable rate of correction between the two groups [19]. These contrasting findings between the Kang study and our results could be explained by the higher size of our cohorts, and by the mean difference between groups in the valgus angle, that was approximately doubled.

Concerning the baseline characteristics, as expected, the two groups differed for demographics, anthropometric features, and radiographic parameters. Interestingly, MO children presented with a more severe radiographic valgus deformity, although the clinical appearance, as measured by IMD, was similar between groups. We highlight that children with idiopathic genu valgum showed a high rate of being overweight (55%) and obese (4%) compared with MO children (and more generally with the Italian pediatric population) [24]. The relationship between overweight and idiopathic genu valgum in children has been reported in previous studies [10,30]; therefore, we recommend a thorough investigation of these patients to rule out endocrinological and/or eating disorders, avoiding possible overtreatment of the physiologic genu valgum.

Idiopathic cases more frequently underwent bilateral femoral stapling at a mean age of 12 years, achieving complete correction or overcorrection in 88% knees, in an average of 14.3 months. These results are consistent with previous studies, reporting complete correction in 70%–100% of idiopathic cases [13,15,19]. 

We found that the preoperative MAD was an independent predictor of the postoperative correction both in MO and idiopathic children. Therefore, we suggest to treat the severe genu valgum earlier or to combine femoral and tibial HeS, increasing the chances of successful correction when the residual growth is insufficient [31]. Nonetheless, this strategy must be corroborated by further studies establishing safety, cost-effectiveness, and potential thresholds for proposing combined femoral and tibial stapling. In our experience, MO children underwent stapling more than one year before the idiopathic counterpart, in line with Boero et al., who recommended earlier stapling in the case of pathologic physes [15]. Despite this, we did not reach complete correction in 45.5% of MO children, and four of them required supplementary corrective surgery (12%). This is a major concern of HeS in MO and it could be explained by several factors. 

First, MO is intrinsically characterized by abnormal growth of the physes, reducing the stature [3,32] and potentially affecting the outcomes of HeS. This is supported by the fact that, in our study, the patient’s height was a predictive factor of the correction rate. Second, the mean age at stapling, which was on average 11 years in our MO cohort, could be too late for achieving complete correction, due to the diminution of expected residual growth; thus, MO children should probably undergo HeS at 8–10 years of age, as recommended by some authors [15,18,31]. Third, MO children exhibited more severe radiographic angular deformity and greater valgus of the tibia, associated with an alteration of the tibio-femoral and tibio-fibular ratio, as already reported [8,33]. This was the reason why the site of stapling was different between the two groups: about two-thirds of MO children underwent hemiepiphyseal tibial stapling, while the idiopathic patients had mostly femoral stapling. In general, the correction must be done where the deformity is, and in MO children the deformity more frequently involves the proximal tibia. However, in MO, the tibia is frequently shorter and associated with hypoplastic fibula, which may impart a tethering effect to the lateral growth of the proximal tibia. Moreover, some studies demonstrated that femoral correction is faster than tibial correction. [17,31]. Both these aspects could explain the lower correction velocity in MO patients; although, in our regression model, the site of stapling could not predict the rate and velocity of correction, probably because of the small sample size.

The present study must be interpreted considering some limitations. The retrospective nature of the study design restricted the availability of data, limiting the sample size. In addition, we lacked a large number of long-standing radiographs at least one year after staple removal, since we could not require X-rays in children simply for research purposes. This limits our understanding of the phenomena that happens when inhibited physes resume growing, in particular, the rebound effect. 

## 5. Conclusions

In conclusion, guided growth by temporary HeS was effective in correcting genu valgum in both idiopathic and MO children, although the latter achieved a lower rate of complete correction, showed slower correction velocity, and reported a higher rate of complications. We suggest to anticipate the timing of stapling and combineg femoral and tibial stapling when the residual growth is insufficient, but further studies assessing safety, cost-effectiveness, and potential thresholds for different treatments are required.

## Figures and Tables

**Figure 1 children-08-00287-f001:**
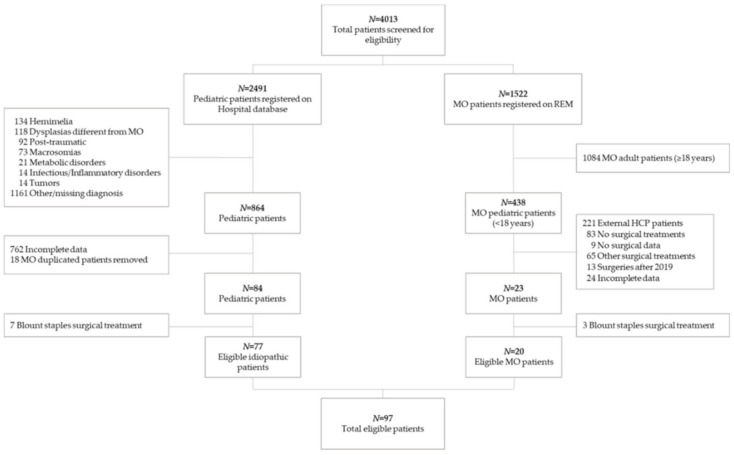
Flow chart of patients included in the study. REM: Registry of Multiple Osteochondromas. MO: multiple osteochondromas. HPC: healthcare provider (i.e., hospital, university, etc.).

**Figure 2 children-08-00287-f002:**
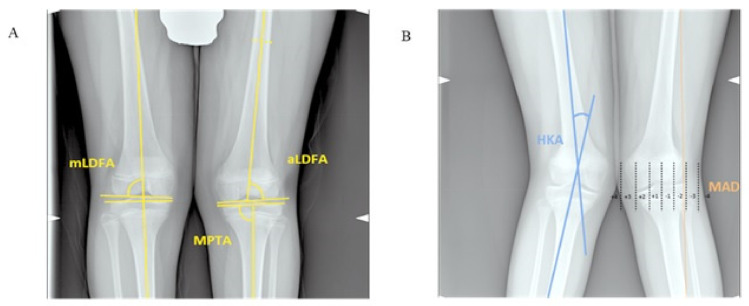
Long-standing anteroposterior radiographs of the lower extremities illustrating the method for determining the (**A**) mechanical lateral distal femoral angle (mLDFA), anatomical lateral distal femoral angle (aLDFA), medial proximal tibial angle (MPTA), (**B**) hip–knee–ankle angle (HKA), and mechanical axis deviation (MAD) in genu valgum. (**A**) The mLDFA and aLDFA are the lateral angle formed between the lines of the mechanical and anatomical femoral axes, respectively, and the articular surface of the distal femur. The MPTA is the medial angle formed between the line of the mechanical tibial axis (that corresponds to the anatomical tibial axis) and a line tangent to the joint surface of the proximal tibial plateau. The HKA is the angle between a line drawn from the center of the femoral head to the center of the knee joint and a line drawn from the center of the knee joint to the center of the tibial plafond. Positive values for varus knee, negative values for valgus knee. The MAD is the perpendicular distance from the mechanical axis of the lower extremity line to the center of the knee joint. MAD zones were labelled from 1 to 4, corresponding to the severity of the deformity. Positive values were assigned for valgus and negative values for varus deviations. Deviations were expressed as a percentage to one-half of the width of the tibial plateau, and the knee joint was divided into four quadrants (two negative and two positive). By definition, deviation between 0% and 50% falls into zone 1, 51% to 100% into zone 2, 101% to 200% into zone 3, and >200% into zone 4.

**Figure 3 children-08-00287-f003:**
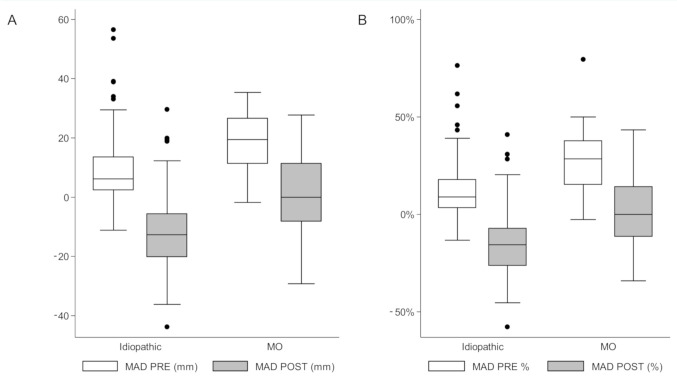
Box plots illustrating preoperative and postoperative mechanical axis deviations (MADs) in MO and idiopathic groups, expressed as absolute and relative deviation. (**A**) MADs expressed as an absolute value. (**B**) MADs expressed as a relative deviation. The black dots indicate the outliers, i.e., observations with values not included within 1.5× IQR (interquartile range) from upper and lower quartiles.

**Figure 4 children-08-00287-f004:**
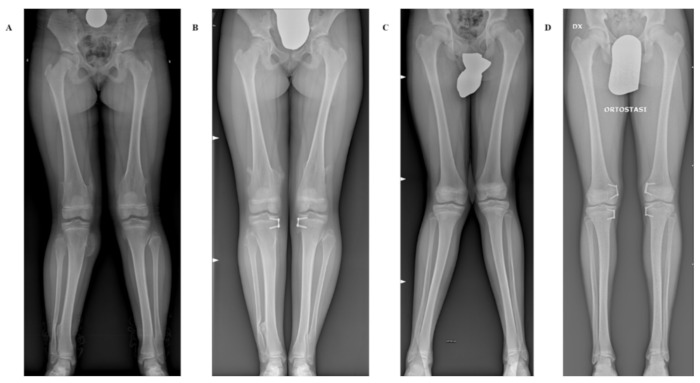
Long-standing anteroposterior radiographs showing two representative cases with and without MO undergoing hemiepiphyseal stapling (HeS). (**A**,**B**) Pre- and postoperative radiographs of a 13-year-old girl affected by genu valgum in MO treated by tibial HeS. (**C**,**D**) Pre- and postoperative radiographs of a 13-year-old boy affected by idiopathic genu valgum treated by combined femoral and tibial HeS.

**Table 1 children-08-00287-t001:** Patients’ characteristics at baseline.

Variables		Idiopathic (*N* = 77)	MO ^a^ (*N* = 20)	*p*-Value *
Gender, *n* (%)				0.098
Female		35 (45)	5 (25)	
Male		42 (55)	15 (75)	
Age, years	Mean (SD ^b^)	12.0 (1.5)	10.8 (1.8)	0.001
Height, cm	Mean (SD)	1.58 (0.10)	141 (0.11)	<0.001
Height (z-score ^c^)	Mean (SD)	0.87 (1.66)	−0.47 (1.09)	<0.001
Weight, kg	Mean (SD)	59(10)	40(9)	<0.001
Weight (z-score)	Mean (SD)	1.13 (0.91)	0.03 (0.95)	<0.001
BMI ^d^	Mean (SD)	23.6 (2.9)	20.1 (3.0)	<0.001
BMI (z-score)	Mean (SD)	1.00 (0.74)	0.31 (0.96)	0.005
IMD ^e^, cm	Mean (SD)	11.6 (2.6)	11.1 (1.4)	0.670
Laterality, *n* (%)				<0.001
Unilateral		3 (4)	7 (35)	
Bilateral		74 (96)	13 (65)	

^a^ MO: multiple osteochondromas; ^b^ SD: standard deviation; ^c^ z-scores were calculated using the Cacciari growth charts [25]; ^d^ BMI: body mass index; ^e^ IMD: inter-malleolar distance. * *p*-values: Mann–Whitney U test (Wilcoxon) for continuous variables, χ^2^ or exact Fisher test for categorical variables.

**Table 2 children-08-00287-t002:** Limbs’ characteristics at baseline evaluation.

Limbs’ Characteristics		Idiopathic (*N* = 151)	MO ^a^ (*N* = 33)	*p*-Value *
Affected side, *n* (%)				0.85
Left		75 (50)	17 (52)	
Right		76 (50)	16 (48)	
**Radiographic Parameters**				
MAD ^b^, mm	Mean (SD)	8.8 (10.6)	17.9 (10.0)	<0.001
MAD, %	Mean (SD)	11.8 (13.7)	27.7 (17.1)	<0.001
aLDFA ^c^, degree (°)	Mean (SD)	79.7 (2.8)	80.5 (4.4)	0.047
mLDFA ^d^, degree (°)	Mean (SD)	85.3 (2.5)	85.8 (3.4))	0.27
MPTA ^e^, degree (°)	Mean (SD)	90.2 (2.8)	95.1 (4.7)	<0.001
HKA ^f^, degree (°)	Mean (SD)	3.4 (3.1)	8.0 (4.1)	<0.001
x-Ray femur, cm	Mean (SD)	48.3 (4.5)	41.4 (4.5)	<0.001
x-Ray tibia, cm	Mean (SD)	38.8 (3.7)	32.3 (3.8)	<0.001
x-Ray fibula, cm	Mean (SD)	38.5 (3.6)	30.8 (3.8)	<0.001
Tibia/fibula ratio	Mean (SD)	1.01 (0.01)	1.05 (0. 04)	<0.001
Femur/tibia ratio	Mean (SD)	1.24 (0.04)	1.28 (0.06)	<0.001
**Surgical Variables**				
Site of stapling, *n* (%)				<0.001
Distal femur		120 (79.2)	8 (24.2)	
Proximal tibia		20 (13.2)	21 (63.6)	
Distal femur and proximal tibia		8 (5.3)	2 (6.1)	
Other ^g^		3 (2)	2 (6.1)	
Duration of correction, month	Mean (SD)	14.3 (5.5)	24.7 (9.2)	<0.001

^a^ MO: multiple osteochondromas; ^b^ MAD: mechanical axis deviation; ^c^ aLDFA: anatomical lateral distal femoral angle; ^d^ mLDFA: mechanical lateral distal femoral angle; ^e^ MPTA: medial proximal tibial angle; ^f^ HKA: hip–knee–ankle angle; ^g^ Other cases included the association of tibial medial hemiepiphysiodesis and femoral distal epiphysiodesis (two limbs) and tibial proximal epiphysiodesis and femoral medial hemiepiphysiodesis (three limbs); * *p*-values: Mann–Whitney U test (Wilcoxon) for continuous variables, χ^2^ or exact Fisher test for categorical variables.

**Table 3 children-08-00287-t003:** Limbs’ characteristics at postoperative evaluation.

Limbs’ Characteristics		Idiopathic (*N* = 151)	MO ^a^ (*N* = 33)	*p*-Value *
**Radiographic Parameters**				
MAD ^b^, mm	Mean (SD)	−12.20(11.89)	−0.13 (14.91)	<0.001
MAD, %	Mean (SD)	−15.36(15.33)	0.28 (19.77)	<0.001
aLDFA ^c^, degree (°)	Mean (SD)	85.6 (4.5)	81.7 (5.3)	<0.001
mLDFA ^d^, degree (°)	Mean (SD)	90.5 (4.2)	86.6 (4.1)	<0.001
MPTA ^e^, degree (°)	Mean (SD)	89.1 (3.4)	90.0 (5.3)	0.46
HKA ^f^, degree (°)	Mean (SD)	−2.75 (3.4)	0.66 (5.21)	<0.001
Correction velocity, mm/month	Mean (SD)	1.66 (0.97)	0.93 (1.10)	<0.001
Achievement of correction, *n* (%)				<0.001
Unresolved		3 (2)	3 (9.1)	
Partially resolved		15 (9.9)	12 (36.4)	
Completely resolved		91 (60.3)	14 (42.4)	
Overcorrected		42 (27.8)	4 (12.1)	
**Complications**				
CDS ^g^ > 2 (major complications), *n* (%)				<0.001
No		148 (98)	23 (70)	
Yes		3 (2)	10 (30)	
CDS > 2 (excluding excision of OCs), *n* (%)				0.006
No		148 (98)	29 (88)	
Yes		3 ^h^ (2)	4 ^i^ (12)	

^a^ MO: multiple osteochondromas; ^b^ MAD: mechanical axis deviation; ^c^ aLDFA: anatomical lateral distal femoral angle; ^d^ mLDFA: mechanical lateral distal femoral angle; ^e^ MPTA: medial proximal tibial angle; ^f^ HKA: hip–knee–ankle angle; ^g^ CDS: Clavien-Dindo-Sink score; ^h^ one tibial overcorrection treated by valgus osteotomy at the end of growth, one deep infection after staple removal treated by surgical debridement, one plate malposition treated by plate repositioning; ^i^ four supplementary staplings for valgus rebound. * *p*-values: Mann–Whitney U test (Wilcoxon) for continuous variables, χ^2^ or exact Fisher test for categorical variables.

## Data Availability

Data available on request due to restrictions. The data presented in this study could be available on request from the corresponding author. The data are not publicly available due to national privacy regulations.

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
