# Peer review of "Outcomes of Temporary Hemiepiphyseal Stapling for Correcting Genu Valgum in Children with Multiple Osteochondromas: A Single Institution Study"

_children, 2021, doi:10.3390/children8040287_

Round 1

Reviewer 1 Report

This article is the largest case series of HeS for Genu valgum correction in MO children. The article sets out to investigate the outcomes of hemiephiphyseal stapling (HeS) for correcting genu valgum in idiopathic and children with multiple osteocondromas (MO). 84 patients were of the idiopathic cohort and 23 MO, with all patient baseline characteristics besides gender and IMD being statistically different between the two groups. Correction was primarily assessed with the absolute mechanical axis deviation (MAD) with location +1 or +2 at staple removal being considered a successful outcome and correction velocity to understand the speed of correction. Complications were classified in a variation of the Clavien-Dindo-Sink system. HeS was successful in correcting 81% of limbs with a significantly higher success rate for the idiopathic group, 87% vs 51%, and faster correction rates. MO and MAD were predictors for successful outcomes or correction velocity.

Strengths 

This article is well organised and contains all components as expected.

Introduction was easy to follow and rationale clear. 

Patient selection and ethics well identified.

The literature is well synthesized.

Major criticisms 

In this comparison study there are 84 in one cohort and 23 in the other. Not only that all baseline patient characteristics are significantly different besides gender and IMD. I understand that this is a retrospective study and the difficulties around methodology, however if the cohorts were to begin with such discrepancies in a comparison study, it would be safe to say that the results would be compromised due to the high possibility of confounding variables and therefore difficult to interpret. Multivariate analysis of variables was conducted, however this was for the group as a whole and the effect within each group was not identified, if such confounders were to be accounted for. In Kang et al’s study, comparisons were conducted with matched patients amongst the idiopathic group. Making such a group for comparison would be of benefit to this study.

I am uncertain of the accuracy and reliability of some of the results and interpretations there after of this paper. Mainly being that MO patients have inferior Genu valgum correction in terms of absolute values, when compared to the idiopathic group. This is mainly due to the above mentioned reason of a lack of a matched cohort and the large baseline discrepancies. In line 240, Kang et al study is mentioned and the authors state that their results differ, but do not precede to explain further on why they believe this is the case. This should be further elaborated on and explanation provided for the difference. In Kang et al, when comparisons were conducted with matched cohorts, comparable Genu valgum deformity correction was found and similar results were found of slower correction rates in MO patients. I wonder if there will be a difference in correction rates between MO and idiopathic patients if this study had a matched cohort to compare the results with. Therefore the statement that "inferior" (title) correction is achieved is jumping too far ahead without a sound scientific basis at this stage. At the minimum, please provide an explanation of the differing results and methodology to papers such as Kang et al or provide a matched cohort and new results that support the conclusion.

Furthermore, the suggestion of a procedures of combining femoral and tibial HeS, especially in the conclusions, is not evidence based, seeing that in matched cohorts comparable rates of correction may possibly be found and like other studies suggest that simply having the operation earlier may suffice. Please add references in the literature that outline the combined technique and its benefits in correcting severe genu valgum, and then the rationale for the suggestion that this will help MO patients with evidence in the literature.

For lines 133- 149, I was confused with whether this was the legends for figure 2 or part of the manuscript. Seeing the structure of A. And B. It seems it would be best to simply include this in the legend, or remove the structure and define the parameters with references to the figure, i.e. Fig 2(A) etc. 

Line 262-265. This paragraph is simply a stating of the results. Please elaborate further to more clearly present how it adds to the development of the discussion and interpretation of results.

Minor criticisms 

Line 50 “the effectiveness of HeS”

Line 51 missing a capital letter 

Line 94 “as a landmark”

Line 103-104 sentence incomplete

Line 150 mention if observers were blinded or not

Line 164 “evaluated”

Line 165 “Due to the reduced ”

Line 168 two spaces before “To”

Line 171 “For sensitive” instead of “As sensitive”

Line 180 Why is this “As expected” - reference 

Line 186 Table 2 was difficult to follow, this may be due to the automatic formatting when the manuscript is generated, however a table like in Kang et al, may be more legible and easy to follow. 

Line 212 Table 3 is missing a legend for the abbreviations 

Line 240-242 This sentence does not make sense 

Line 245 “presented with more”

Line 246 “of being overweight”

Line 254 “in an average of 15 months”

Line 267 “in line with a previous”

Line 274 “undergo HeS”

Line 276 “of the tibia”

Line 280-283 When accounting for the reasons why failure occurred, it is stated that site may be a reason however it is immediately rebutted with how site did not matter in line 281-283. Restructuring of these sentences is advised.

Line 286 “number” instead of “part”

Line 287 Advise to not use “just” — rather use “simply"

Line 288 “that happens when”

Author Response

Dear Reviewer,

thank you for giving us the opportunity to revise the paper. Your comments were insightful and enabled us to improve the quality of our manuscript. According to your suggestions, we rewrite contents of all the sections.  

Major criticisms
In this comparison study there are 84 in one cohort and 23 in the other. Not only that all baseline patient characteristics are significantly different besides gender and IMD. I understand that this is a retrospective study and the difficulties around methodology, however if the cohorts were to begin with such discrepancies in a comparison study, it would be safe to say that the results would be compromised due to the high possibility of confounding variables and therefore difficult to interpret. Multivariate analysis of variables was conducted, however this was for the group as a whole and the effect within each group
was not identified, if such confounders were to be accounted for. In Kang et al’s study, comparisons were conducted with matched patients amongst the idiopathic group. Making such a group for comparison would be of benefit to this study.

Following your suggestions, we have performed integrative analyses using propensity score approaches (in line with Kang et al.). The propensity score was used both as covariate and as matching factor leading to results, that are consistent with multivariable analysis findings. Due to the reduced sample size for the propensity score matching with a caliper of 0.40 SD (N=22), the difference between the two cohorts was not statistically significant for MAD post, although the point estimate of mean difference (15.96) was nearly identical to multivariable results.

We have included a sentence on this comparison into statistical analyses and results and we have drafted a table showing the results (supplementary table 5).

I am uncertain of the accuracy and reliability of some of the results and interpretations there after of this paper. Mainly being that MO patients have inferior Genu valgum correction in terms of absolute values, when compared to the idiopathic group. This is mainly due to the above mentioned reason of a lack of a
matched cohort and the large baseline discrepancies.

You are correct, MO patients do not have inferior genu valgum correction in terms of absolute values. As you can see the effect of stapling in terms of correction was comparable between the two groups. However, MO children achieved partial correction in almost half of cases, mainly because they had a more severe valgus at baseline. As mentioned before the matching reduced drastically the size of the cohort and consequently the statistical significance of our findings.

In line 240, Kang et al study is mentioned and the authors state that their results differ, but do not precede to explain further on why they believe this is the case. This should be further elaborated on and explanation provided for the difference. In Kang et al,
when comparisons were conducted with matched cohorts, comparable Genu valgum deformity correction was found and similar results were found of slower correction rates in MO patients. I wonder if there will be a difference in correction rates between MO and idiopathic patients if this study had a matched cohort to compare the results with. Therefore the statement that "inferior" (title) correction is achieved is jumping too far ahead without a sound scientific basis at this stage. At the minimum,
please provide an explanation of the differing results and methodology to papers such as Kang et al or provide a matched cohort and new results that support the conclusion.

Accordingly, we have revised the title, sentence (starting at line 240), providing further explanations. Please also consider our previous reply to your comment, that supports our conclusions (supplementary table 5).

Furthermore, the suggestion of a procedures of combining femoral and tibial HeS, especially in the conclusions, is not evidence based, seeing that in matched cohorts comparable rates of correction may possibly be found and like other studies suggest that simply having the operation earlier may suffice. Please add references in the literature that outline the combined technique and its benefits in correcting severe genu valgum, and then the rationale for the suggestion that this will help MO patients with evidence in the literature.

Following your suggestion, we found a reference (Dai ZZ et al. 2021) that shows a notable success rate (97%) for combined femoral and tibial HeS, that is higher than tibial HeS (88%) and femoral (95%). We have cited the reference in the discussion section and consequently added it into reference list. 

For lines 133- 149, I was confused with whether this was the legends for figure 2 or part of the manuscript. Seeing the structure of A. And B. It seems it would be best to simply include this in the legend, or remove the structure and define the parameters with references to the figure, i.e. Fig 2(A) etc.

The section represents the legend of figures 2A and 2B. We have drafted it following the journal guidelines for figures, but we agree with you that it is a bit confusing. To simplify, we have added a couple of rows to separate the legend text from the results section.

Line 262-265. This paragraph is simply a stating of the results. Please elaborate further to more clearly present how it adds to the development of the discussion and interpretation of results.

We agree completely with your comment. We have moved the sentence into results, and add a short dissertation

Minor criticisms

Following your fruitful suggestions, we have corrected the typos, simplify Table 2 (by dividing in 2 tables) and rewrite the sentences that you have highlighted to improve the content.

Reviewer 2 Report

Overall a good article. Intersting topic for this reviewer.

The study design is clear and there seem to bee no mistakes in the numbers.

Comments:

Abstract: small comment. I would advise the term guided growth or guided correction for the hemi-epiphysiodesis technique.

Introduction: small comment. Line 40. osteochondroma arise , mainly from the long bones, but also in other places.

Material and Merthods: I understand staples and 8-plates are used. On page 5 line 196, I understand that 90% was 8-plates. How was this divided among the groups. They are different technique with different cerrection velocity. Why are they combined. is it not better to leave the staple group out of this study? 

In the introduction is written that 38% of MO patients have valgus deformities around the knees.  In the end 23 patients are analysed that is only 5,2 % of the 438 cases, Why are External patients included in figure one. Without them 38% would still be 82 patients. 23 in not a lot and very few for good statistics.

Results: Table 2 is a bit long. I would advise to make is smaller and put less details in it. Or divide it in 2 Tables. Table 3 is very thechnical en difficult to understand for me. 

How were all the complications adressed. In the infection case; did the 8-plate needed to be removed?

Discussion: add comment or explanation about the correction velocity.

Is it just the MO or also the location of the 8-plates. Correction velocity is faster in the femur then in the tibia, is this the difference in velocity also reason why the idiopathic group goes faster.

In general the correction must be done were the deformity is. In MO patients it can happen that you correct the tibia and the a deformity comes in the femur. Was this seen in this group. This could be a reason the treat the femur and tibia at the same time.

In MO patients sometimes the deformities comes at a very young age. Then hemi-epifysiodesis is performed at around 7 years of age. The you will always see a relapse later in live. Wasd this seen in this serie?

Was the velocity of the correction used for calculating the age of the hemi-epifysiodesis? It also depends on growth spurts, how do you adress that?

Author Response

Dear Reviewer,

thank you for giving us the opportunity to revise the paper. Your comments were insightful and enabled us to improve the quality of our manuscript. According to your suggestions, we rewrite contents of all the sections. Abstract: small comment. I would advise the term guided growth or guided correction for the hemi-epiphysiodesis technique As you suggested, we have introduced guided growth into the abstract and introduction, keeping also “HeS” as acronym.  Introduction: small comment. Line 40. osteochondroma arise, mainly from the long bones, but also in other places.

We have slightly changed the sentence.

Material and Methods: I understand staples and 8-plates are used. On page 5 line 196, I understand that 90% was 8-plates. How was this divided among the groups. They are different technique with different correction velocity. Why are they combined. is it not better to leave the staple group out of this study?

We agree with your observations. We included just 10% of patients with staples, while the majority of cases underwent HeS using 8-plates. We decided to exclude the group with staples from the present study, following your suggestion. The statistical analysis was repeated, and the manuscript was revised accordingly.

In the introduction is written that 38% of MO patients have valgus deformities around the knees. In the end 23 patients are analysed that is only 5,2 % of the 438 cases, Why are External patients included in figure one. Without them 38% would still be 82 patients. 23 in not a lot and very few for good statistics.

We thank the reviewer for this comment. As reported in the introduction, Nawata et al. found that knee valgus can be present up to 38% of patients with MO. This is an epidemiological data that is confirmed by other studies. The flowchart explain how patients were included in the study. As explained in the methods, data were extracted from two different sources: the hospital database, that includes patients admitted to the Unit of Pediatric Orthopedics, and from the Registry of Multiple Osteochondromas which is located at the same hospital but collects data also from other (external) institutions. We have integrated this detail into the Materials and Methods. The figure was revised accordingly.

Results: Table 2 is a bit long. I would advise to make is smaller and put less details in it. Or divide it in 2 Tables. Table 3 is very technical and difficult to understand for me.

Following your comment, we have divided Table 2 in 2 separate tables ù: one for baseline data, the second for post-operative data. We have renamed tables accordingly.

In addition, table 3 (now supplementary table 4) was removed from the results, to simplify the understanding, following your input.

How were all the complications addressed. In the infection case; did the 8-plate needed to be removed?

Major complications in MO children included supplementary stapling for valgus rebound (4/184) and surgery for recurrent OCs around the knee (6/184), while major complications requiring further surgical interventions in idiopathic patients included: one deep infection following staple removal, treated by surgical debridement and 2-weeks antibiotic therapy; one plate malpositioning, treated by plate repositioning 2 months later the first operation; one tibial overcorrection requiring valgus osteotomy at the end of growth.

Discussion: add comment or explanation about the correction velocity. Is it just the MO or also the location of the 8-plates. Correction velocity is faster in the femur then in the tibia, is this the difference in velocity also reason why the idiopathic group goes faster.

We thank you for your comment. We acknowledge that correction velocity is faster in the femur then in the tibia as reported by several authors (Dai ZZ et al. 2021). The site of stapling was not a significant predictor of correction in our regression model maybe due to small sample size and predominant effect of other factors, such as MO disease, gender, and pre-operative deformity. We updated the discussion accordingly  

In general the correction must be done were the deformity is. In MO patients it can happen that you correct the tibia and the a deformity comes in the femur. Was this seen in this group. This could be a reason the treat the femur and tibia at the same time.

We agree with your comment. The correction must be done where the deformity is. Unfortunaly, in MO patients the deformity mainly involves the proximal tibia, this may partially explain our inferior outcomes in MO cohort.

Concerning the second point of the question, we did not observe this phenomenon in our MO patients. Generally, in our practice, the main reason for proposing combined femoral and tibial stapling was to achieve complete correction in case of severe valgus deformity in children with low residual growth. In our experience, this occurred in about one out of ten cases, thus not being sufficient to propose an evidence-based recommendation. Further studies are required to establish safety, cost-effectiveness, and potential thresholds for proposing combined femoral and tibial stapling. We revised the discussion accordingly.

In MO patients sometimes the deformities come at a very young age. Then hemi-epiphysiodesis is performed at around 7 years of age. The you will always see a relapse later in live. Was this seen in this series?

In our population, only 3 children were treated younger than 8 years. Of them, 1 case with MO was treated at 5 .5 years of age and underwent relapse of the deformity after staple removal, when he was 11. We acknowledge that anticipating the stapling at a very young age could produce an increased rebound effect, this aspect is debated in literature [Dai ZZ et al. 2021]. Nonetheless, with our numbers the age at stapling was not a significant predictor of relapse.

Was the velocity of the correction used for calculating the age of the hemi-epiphysiodesis? It also depends on growth spurts; how do you address that?

We thank the reviewer. Unfortunately, we cannot answer this question since the study was retrospective, cohort was treated by more than ten different surgeons and we lack complete information from health records, concerning the surgical decision making. Generally, in our practice the HeS is proposed after age of 8 since it is still possible to observe spontaneous correction of the valgus deformity especially in idiopathic cases. The mean age of stapling in our cohort was 12 years, in line with other studies [Boero et al. 2011, Wiemann et al. 2009, Kang et al. 2017].

Round 2

Reviewer 1 Report

Thank you for your revisions. 

It was a pleasure to read your work.

There are minor grammatical errors here and there but the paper otherwise looks good for publication. It would be of benefit if a fluent English speaker was to quickly edit some of the grammar so it reads more smoothly. 

Thank you